# OmniMotion: Multimodal Motion Generation with Continuous Masked Autoregression

## Abstract

Whole-body multi-modal human motion generation poses two primary challenges: creating an effective motion generation mechanism and integrating various modalities, such as text, speech, and music, into a cohesive framework. Unlike previous methods that usually employ discrete masked modeling or autoregressive modeling, we develop a continuous masked autoregressive motion transformer, where a causal attention is performed considering the sequential nature within the human motion. Within this transformer, we introduce a gated linear attention and an RMSNorm module, which drive the transformer to pay attention to the key actions and suppress the instability caused by either the abnormal movements or the heterogeneous distributions within multi-modalities. To further enhance both the motion generation and the multimodal generalization, we employ the DiT structure to diffuse the conditions from the transformer towards the targets. To fuse different modalities, AdaLN and cross-attention are leveraged to inject the text, speech, and music signals. Experimental results demonstrate that our framework outperforms previous methods across all modalities, including text-to-motion, speech-to-gesture, and music-to-dance. The code of our method will be made public.

## 1 Introduction

Whole-body human motion generation represents an expanding frontier in computer vision, offering significant value across a variety of applications, including film production, gaming, virtual reality, robotics, and so on. Broadly speaking, motion generation could be conditioned on various signals, such as text, speech, music, and more.

Historically, approaches to whole-body motion generation usually focus on isolated tasks. Typically, they either address text-to-motion generation, or concentrate on speech-to-gesture translation, or engage in music-to-dance synthesis. Despite their successes in single task, their frameworks are exclusively designed for individual tasks and cannot be easily adapted to different tasks. In addition, they tend to overlook the underlying commonalities that exist across different tasks. In contrast, in this work we seek to address these motion generation challenges from various signals within an omni-framework. This brings two advantages: 1) It allows each modality to benefit from the patterns present in other modalities, preventing single-mode solutions from becoming trapped in a local minimum; 2) It enhances each task with data from other tasks, which is particularly relevant given the limited scale of data available for individual motion tasks.

Previous studies in motion generation generally proceed in two paths. The first employs the vector quantization (VQ) technique to convert continuous motion to discrete tokens, and then performs autoregressive or masked modeling to predict the tokens (Zhang et al., 2023d;a; Kong et al., 2023; Zhong et al., 2023; Guo et al., 2024). While this path effectively utilizes the strengths of autoregressive and masked modeling, the quantization step inevitably introduces approximation errors, which impose undesirable limits on the quality of the generated motions. The second directly regresses the continuous motions using techniques such as generative adversarial networks (GANs) (Tulyakov et al., 2018), variational autoencoders (VAEs) (Xu et al., 2020; Ahuja & Morency, 2019; Petrovich et al., 2022; Guo et al., 2022a), or recent diffusion models (Chen et al., 2023; Tevet et al., 2023; Zhang et al., 2024b; 2023b; Ribeiro-Gomes et al., 2024). Despite avoiding the approximation errors, they miss the autoregressive or masked modeling technologies, which have been shown to deliver superior performance in motion generation tasks. Consequently, the performance of the motion generated by this path is overall lower than that achieved by the first path.

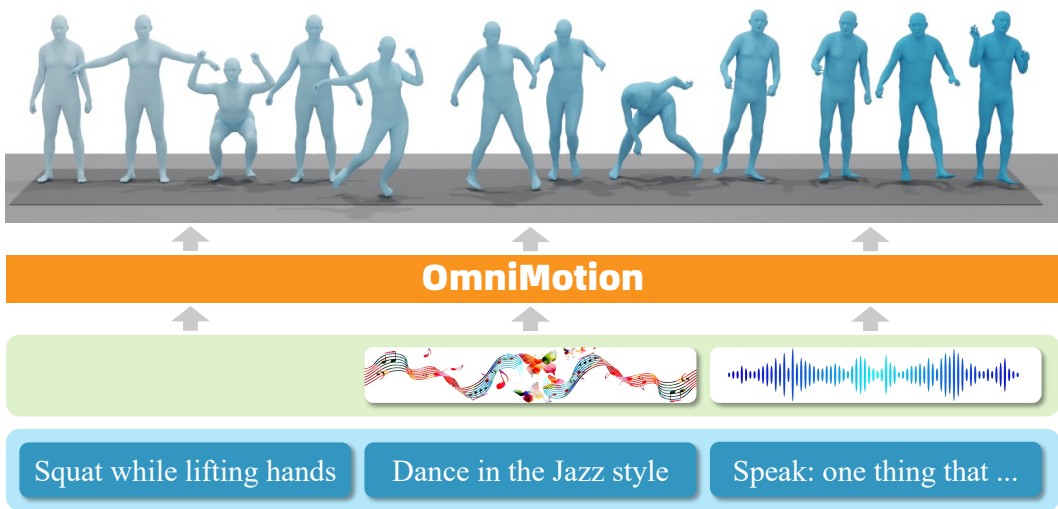

Figure 1: We construct an omni motion framework with a continuous masked autoregressive motion transformer for multimodal whole-body motion modeling, including text-based, music-based, and speech-based motion generation.

To both leverage the advantages of autoregressive and masked modeling, and the benefits of the continuous motion representation, in this work we combine them together to propose a continuous masked autoregressive motion generation framework. We apply a random masking on the sequential motion tokens, and employ a transformer to autoregressively predict the masked tokens. Unlike the visual MAR (Li et al., 2024b), we sequentially predict the masked tokens with causal attention rather than performing random reordering, considering the sequential nature within human motion. To enhance the MAR modeling in motion space, we introduce a gated linear mechanism and an RMSNorm module. The gated linear mechanism serves as an adaptive feature selector, driving the transformer to not only pay more attention to the key actions, like gesture switching and large movement, but also disregard less relevant frames and suppress redundant actions, like stationary motions. The RMSNorm is particularly advantageous in scenarios with features exhibiting a large dynamic range, e.g., our unified framework for multi-modalities, where the input distributions are highly heterogeneous. In addition, RMSNorm helps relieve the gradient instability caused by the abnormally large motions, such as sudden jumping or turning back. After the masked autoregressive transformer, the calculated attention of the masked tokens is fed into a series of DiT blocks to diffuse towards the target tokens, which are decoded to the generated motions.

In addition to the text-based motion generation, we further extend our framework to multimodal conditions. Building upon a similar structure, the multimodal signals are fused by AdaLN (Guo et al., 2022c) and cross-attention modules. Extensive experiments across different datasets demonstrate our framework can work well with different modalities, including text, speech, and music, and outperform previous methods in whole-body text-to-motion, speech-to-gesture, and music-to-dance tasks.

The main contributions of this work are then summarized as follows:

- We design an omni motion framework for whole-body human motion generation, where one framework encompasses multiple modalities.
- We propose a continuous autoregressive motion transformer with causal attention, where a gated linear mechanism and an RMSNorm module are developed to assist the motion modeling, and the DiT blocks are employed to improve the quality of the generated motions.
- We integrate the multimodal signals via AdaLN and cross-attention, obtaining superior performance than previous methods in text-based, speech-based, and music-based motion generation.

## 2 RELATED WORK

**Text-based Motion Generation.** Previous research on text-based motion generation can be generally categorized into two mainstream paths: continuous regression and discrete classification. In the

continuous regression domain, numerous strategies have leveraged the variational autoencoder (VAE) framework, integrating latent embeddings of encoded text with those of encoded poses, which are then decoded into motion predictions (Xu et al., 2020; Ahuja & Morency, 2019; Athanasiou et al., 2022; Petrovich et al., 2022; Guo et al., 2022a). Other methods have investigated the potential of recurrent networks (Lin et al., 2018; Plappert et al., 2018; Zhang et al., 2020), generative adversarial networks (Cai et al., 2018; Wang et al., 2020; Tulyakov et al., 2018), or transformer networks (Tevet et al., 2022; Lin et al., 2023b; Bhattacharya et al., 2021; Petrovich et al., 2021) to enhance the motion regression quality. Building on the success of diffusion models, recent approaches have begun to integrate the diffusion process into motion diffusion (Kim et al., 2023; Chen et al., 2023; Tevet et al., 2023; Zhang et al., 2024b; Dabral et al., 2023; Lou et al., 2023; Zhang et al., 2023b; Ribeiro-Gomes et al., 2024; Yuan et al., 2023; Wang et al., 2023; Zhang et al., 2023c; 2024d; Bian et al., 2025), yielding impressive results. In the discrete classification domain, the input motion undergoes initial encoding via a VQ-VAE (Van Den Oord et al., 2017), producing motion tokens for subsequent prediction (Zhong et al., 2023; Li et al., 2024d). Drawing inspiration from advancements in natural language processing, some methods utilize autoregressive modeling to predict tokens sequentially (Guo et al., 2022b; Lou et al., 2023; Zou et al., 2024). Others employ generative masked modeling strategies, with tokens randomly masked during training for the model to predict (Guo et al., 2024; Pinyoanuntapong et al., 2024; Yuan et al., 2024). More recently, large language models (LLMs) have been harnessed to help the prediction process, considering their large-scale pretraining (Zhang et al., 2023d; Zhou et al., 2024). In this work, we seek to integrate the most effective elements from these two paths: the continuous diffusion and the masked autoregressive modeling. A previous attempt in this direction (Meng et al., 2024) directly transfers the MAR in image generation (Li et al., 2024b) into motion generation without considering the difference between image and motion spaces, especially the temporal correlation. Also, its framework is only designed for body-only motion generation. Differently, we propose a new MAR mechanism that is especially designed for whole-body motion generation.

**Multimodal Motion Generation.** In addition to text, there are many other signals that various human motions are conditioned on, such as speech and music. In the realm of speech-to-gesture generation, both continuous regression and discrete classification paths have been explored. In the continuous domain, methods employ deep generative models like GANs (Ahuja et al., 2022), normalizing flows (Tan et al., 2024), and diffusion models (Alexanderson et al., 2023; Chen et al., 2024a; He et al., 2024; Yang et al., 2023; Zhu et al., 2023; Chen et al., 2024b) to learn complex motion distributions in the speech data. In the discrete domain, methods leverage either the autoregressive modeling (Yi et al., 2023) or the masked modeling (Liu et al., 2024a;b) to predict the discrete tokens quantified by the VQ-VAE. The primary distinction among these methods lies in their specific handling of different parts of human motion, including body movements, hand gestures, and facial expressions. Similarly, in the realm of music-to-dance generation, there are also methods in both the continuous domain (Zhuang et al., 2022; Tseng et al., 2023; Li et al., 2024a; Huang et al., 2024; Zhang et al., 2024a; Li et al., 2023) and the discrete domain (Siyao et al., 2022). The discrete autoregressive model is leveraged after the motion quantization with VQ-VAE (Siyao et al., 2022). More methods harness the diffusion model to directly regress the target dancing motion in the continuous space (Tseng et al., 2023; Li et al., 2024a; Huang et al., 2024; Li et al., 2023). Recent methods also start to merge autoregressive and diffusion models, producing coherent and music-aligned dance sequences (Zhang et al., 2024a).

Recent works have begun to seek multimodal solutions, i.e., designing one framework for motion generation from different input signals (Luo et al., 2024; Zhang et al., 2024c; Zhou & Wang, 2023; Zhou et al., 2023; Ling et al., 2023; Bian et al., 2025; Li et al., 2024c). Some methods in the discrete domain attempt to incorporate quantized condition tokens into the vocabulary of the generation model (Luo et al., 2024; Zhou & Wang, 2023; Zhou et al., 2023; Zhang et al., 2025), while some methods in the continuous domain try to integrate the multimodal signals by designing the motion ControlNet (Ling et al., 2023; Bian et al., 2025), where the multimodal conditions guide the sampling of a pretrained text-to-motion diffusion model. However, most previous methods are restricted by the varying motion data of different modalities, limiting multi-modal frameworks primarily to body-only motion generation. To overcome this, MotionCraft (Bian et al., 2025) standardizes datasets across modalities into a unified whole-body format that includes body, hands, and face. In this work, we follow this unified representation to build a whole-body multi-modal motion framework, taking advantage of continuous masked auto-regression.

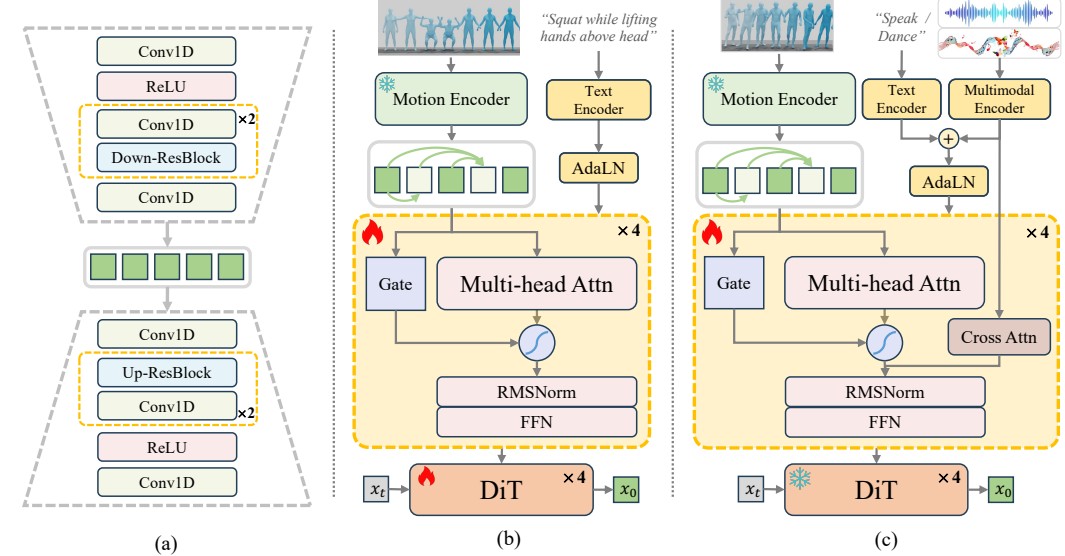

Figure 2: Framework overview. Our framework consists of three parts: (a) The input motion is encoded by an autoencoder to extract a latent code, producing the continuous motion tokens. (b) The motion tokens are masked and predicted in an autoregressive transformer with causal attention, producing conditions for DiTs to diffuse towards the target tokens. (c) Multimodal signals are encoded and then injected via AdaLN and cross-attention.

## 3 METHOD

### 3.1 OVERVIEW

The overview of our framework is illustrated in Figure 2, which is divided into three main stages: In the first stage, we encode the input motion with an autoencoder, which generates continuous motion tokens. In the second stage, we focus on the text-based motion generation, utilizing our motion masked autoregressive framework to model the motion generation process. In this stage, an autoregressive transformer is employed to predict the masked tokens, within which a gated linear mechanism is designed, and an RMSNorm module is employed. The text information is integrated into the transformer via AdaLN after encoding. After the transformer, the generated embedding is fed into the DiT modules as the condition to diffuse towards the target token. In the third stage, we extend the model learned in the previous stage to the multi-modal structure. This involves merging the text embedding with multimodal signal embeddings—specifically speech or music—prior to their AdaLN input. Furthermore, we inject the multimodal embedding through a cross-attention module into the masked transformer. In the multimodal learning stage, the DiT module is kept in the same structure and frozen.

### 3.2 CONTINUOUS AUTOENCODER

To feed the human motion into the transformer, we start by encoding the original motion into motion tokens. Given a motion sequence $\{\mathbf{M}_t\}_{t=1}^T$, the objective of an autoencoder is to extract a latent code $z_{\mathrm{AE}}$ that optimally captures the essence of the original motion. Different from most previous motion generation methods with autoregressive transformers, we use a continuous autoencoder rather than the VQ-VAE to do the encoding, which avoids the precision loss associated with the quantization approximation. In the encoder, we stack the 1D convolution networks with ReLU activation to do the feature processing. Following this, two down-sampling residual blocks are applied to reduce the motion feature size to one-fourth of its original dimensions. In the decoder, the same structure in the reversed order is utilized to up-sample the motion feature back to the original size, producing $\{\hat{\mathbf{M}}_t\}_{t=1}^T$. Therefore, the loss for training the autoencoder is defined as

$$\mathcal{L}_{\mathrm{AE}} = \sum_t \| \hat{\mathbf{M}}_t - \mathbf{M}_t \|_1. \tag{1}$$

The latent code $z_{\mathrm{AE}}$ between the encoder and the decoder serves as the motion tokens $\mathbf{x}^0, \mathbf{x}^1, ..., \mathbf{x}^N$, with a sequence length that is one-fourth of the initial sequence length.

### 3.3 Continuous Masked Autoregression

With the continuous motion tokens, we design a masked autoregressive transformer to model the motion generation, effectively capturing the temporal relationship between different tokens and producing the rich contextual condition $\mathbf{z}^i$ for the subsequent diffusion process. We first randomly mask the motion tokens following language models (Devlin et al., 2018), obtaining some masked tokens $\{\tilde{\mathbf{x}}^i\}$. The temporal masking strategies adopt the same mask ratio schedule following (Chang et al., 2022), and are computed as

$$\gamma(\tau) = \cos(\frac{\pi\tau}{2}), \tag{2}$$

where $\tau \in [0, 1]$. In training, $\tau \sim \mathbf{U}(0, 1)$ is uniformly sampled, leading to a mask ratio $\gamma(\tau)$. Then according to this ratio, $\gamma(\tau) \times N$ tokens are randomly selected to be masked.

After the masking, unlike previous MAR methods (Li et al., 2024b; Meng et al., 2024), our approach does not involve random rearranging of tokens or batch-tokens prediction. Also, we do not perform bidirectional attention. In contrast, we maintain the temporal order of the original motion, and sequentially undertake autoregressive prediction, thus forming a causal attention manner, as illustrated in Figure 2.

For the input text prompt $T$, we first employ the LaMP (Li et al., 2024d) text transformer to extract textual features, leveraging its advanced capabilities to encode the linguistic nuances and semantic structure of the input prompt. This creates a high-dimensional feature representation that is crucial for guiding motion generation process.

Following the feature extraction, we utilize AdaLN to seamlessly integrate the text-derived control signals into the masked autoregressive transformer. AdaLN offers a dynamic approach to normalization, allowing the modulation of its parameters in response to the specific text input, thereby facilitating subsequent multimodal condition injection. By employing this method, we enhance the model's ability to incorporate the guiding signals from the text and other signals into the motion generation process, ensuring that the transformer's output features are better aligned with the intended motion generation goals. The features outputted by the transformer embody a strong directive capacity for motion generation. This enables the model not only to faithfully interpret the semantic content of the input text but also to produce motion sequences that are coherent with and reflective of the textual intent. The enriched output features contribute to achieving smoother transitions and logically consistent motion sequences in complex generation scenarios.

**Gated Linear Mechanism.** We employ a gated linear attention mechanism within the transformer to regulate the attention weights at each time step. Specifically, we compute a gating signal by applying a linear transformation $g_o$ to the input $x$ followed by a sigmoid activation function. This gating signal acts as a dynamic filter, adjusting the output of the attention module based on the relevance of the input features. Consequently, during the attention computation, the final output $o$ is modulated by this gating signal, enabling the model to selectively focus on the most pertinent action frames.

$$o = g \times \text{Softmax}(\frac{Q \cdot K^T}{d_k})V, g = \text{sigmoid}(g_o(x)). \tag{3}$$

This mechanism effectively serves as an adaptive feature selector, allowing the model to disregard less relevant frames and suppress redundant action frames (such as stationary or repetitive motions), thereby enhancing its attention to key actions (e.g., gesture transitions and changes in motion direction). Furthermore, by dynamically adjusting the attention distribution through gating, the model is capable of selectively retaining historical frames or predicting future frames based on the current action state.

**RMSNorm.** Our objective is to construct a unified model that can simultaneously perform text-to-motion, speech-to-gesture, and music-to-dance tasks. Therefore, we aim to ensure that during the fine-tuning process across different datasets, the model does not suffer from instability that could lead to catastrophic failures. RMSNorm (Zhang & Sennrich, 2019) is particularly advantageous in scenarios with features exhibiting a large dynamic range, especially in tasks where the input distributions are highly heterogeneous. This characteristic enables the model to maintain stability when faced with diverse types of inputs or when observing uneven feature distributions. Additionally, RMSNorm has the potential to mitigate the gradient instability that may arise from significant motion variations, such as sudden jumps or rapid turns.

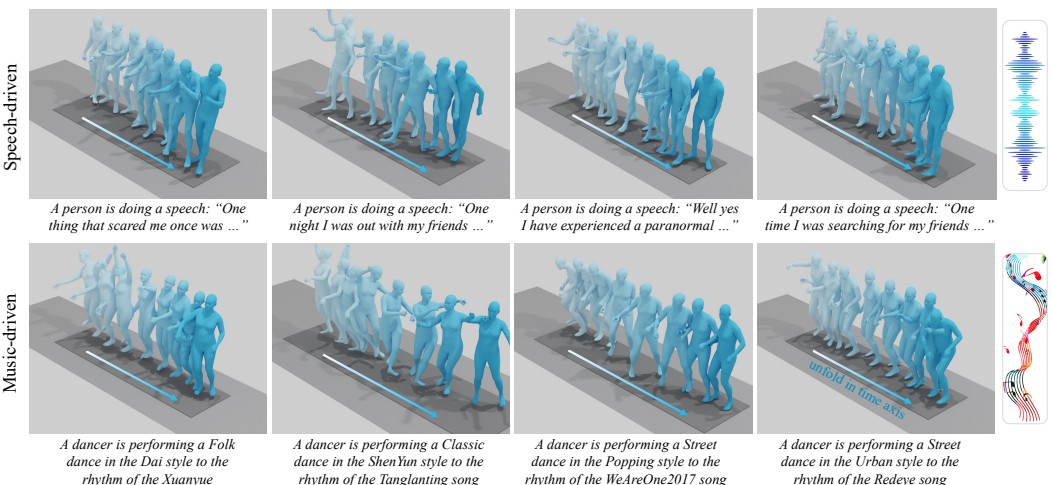

Figure 3: The qualitative results of motions generated from our model driven by speech and music.

### 3.4 DIFFUSION TRANSFORMER

In contrast to previous works (Li et al., 2024b; Meng et al., 2024), we adopt Diffusion Transformer (DiT) as our diffusion model. While the use of DiT may incur additional time costs during training and inference (not much due to the compact structure of motion data), it significantly enhances the quality of generated outputs. Compared to MLPs, DiT provides greater convenience in injecting conditional control signals. During the training process of multimodal generation, we freeze the diffusion model and only fine-tune the masked transformer. The structural characteristics of DiT facilitate this approach, enabling it to better handle various types of conditional signals.

Moreover, MLPs exhibit notable limitations when processing heterogeneous data. This incapacity results in suboptimal performance when confronted with diverse signal types, such as speech and music. Due to the relatively small number of parameters in MLPs, they are prone to overfitting on specific datasets (e.g., text-to-motion). This situation can be analogized to a dancer who is adept only in a single dance style; when asked to incorporate other styles, they appear clumsy and ineffective. Consequently, when we attempt to fine-tune MLPs on a different dataset, they are ill-equipped to adapt to the challenges posed by new signals, leading to failures in multimodal generation tasks.

In contrast, DiT demonstrates superior performance in complex multimodal generation contexts. Its enhanced generalization capabilities allow it to flexibly handle a variety of input signal types, rather than being confined to a single data format. This ensures that the model exhibits increased adaptability and reliability when exposed to diverse data, ultimately resulting in higher-quality outcomes.

### 3.5 TEXT-TO-MOTION PRETRAINING AND MULTIMODAL CONTROL ADAPTATION

We first pre-train the model on text-motion paired data in a text-to-motion generation setting. Owing to its strong semantic expressiveness and cross-modal alignment properties, we adopt text as a shared conditional signal across diverse unimodal datasets, enabling the model to learn sequence-level generation capabilities between text and motion, as well as a coarse-grained textual guidance mechanism for generative control.

We hypothesize that the contextual features output by the masked transformer provide a more expressive control signal compared to raw text embeddings. Accordingly, within the DiT architecture, we inject the transformer's output features by summing them with the time embeddings, thereby guiding the motion generation process as:

$$\tilde{\mathbf{x}}_{t-1}^i \sim p(\tilde{\mathbf{x}}_{t-1}^i | \tilde{\mathbf{x}}_t^i, t + \mathbf{z}^i). \tag{4}$$

Then the training objective for noise prediction is defined as:

$$\mathcal{L} = \mathbb{E}_{\epsilon,t} \| \epsilon - \epsilon_\theta(\tilde{\mathbf{x}}_t^i | t + \mathbf{z}^i) \|. \tag{5}$$

This procedure yields the trained model $\mathcal{M}_{t2m}$. When incorporating additional control signals, we initialize the entire model with the parameters of model $\mathcal{M}_{t2m}$, freeze the DiT, and fine-tune only the

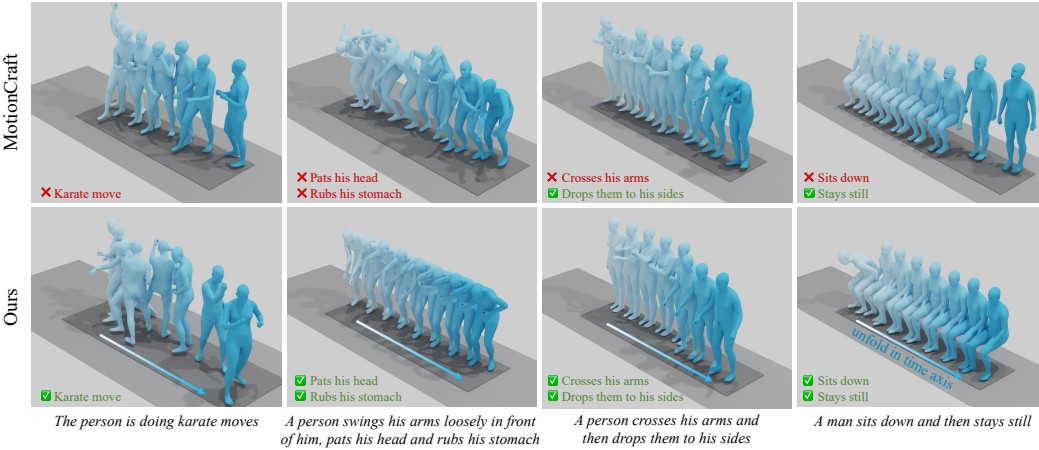

Figure 4: The qualitative results of text-driven motion generation.

| Method | R Precision | | | FID ↓ | MM Dist↓ |
|---|---|---|---|---|---|
| | Top-1 ↑ | Top-2 ↑ | Top-3 ↑ | | |
| GT | $0.663^{\pm0.006}$ | $0.807^{\pm0.002}$ | $0.864^{\pm0.002}$ | $0.000^{\pm0.000}$ | $15.567^{\pm0.036}$ |
| T2M-GPT(Zhang et al., 2023a) | $0.529^{\pm0.004}$ | $0.652^{\pm0.003}$ | $0.732^{\pm0.003}$ | $10.457^{\pm0.108}$ | $17.029^{\pm0.039}$ |
| MDM(Tevet et al., 2023) | $0.383^{\pm0.010}$ | $0.527^{\pm0.012}$ | $0.604^{\pm0.009}$ | $18.671^{\pm0.370}$ | $18.785^{\pm0.054}$ |
| MotionDiffuse(Zhang et al., 2024b) | $0.525^{\pm0.004}$ | $0.675^{\pm0.009}$ | $0.743^{\pm0.009}$ | $9.982^{\pm0.379}$ | $17.314^{\pm0.066}$ |
| FineMoGen(Zhang et al., 2023c) | $0.565^{\pm0.001}$ | $0.710^{\pm0.004}$ | $0.775^{\pm0.004}$ | $7.323^{\pm0.143}$ | $16.679^{\pm0.029}$ |
| MCM(Ling et al., 2023) | $0.407^{\pm0.002}$ | $0.559^{\pm0.003}$ | $0.636^{\pm0.001}$ | $15.540^{\pm0.443}$ | $18.673^{\pm0.029}$ |
| MotionCraft (Bian et al., 2025) | $0.590^{\pm0.003}$ | $0.743^{\pm0.004}$ | $0.804^{\pm0.004}$ | $8.477^{\pm0.102}$ | $16.252^{\pm0.035}$ |
| Ours | $\mathbf{0.704}^{\pm0.003}$ | $\mathbf{0.843}^{\pm0.005}$ | $\mathbf{0.898}^{\pm0.005}$ | $\mathbf{4.838}^{\pm0.100}$ | $\mathbf{15.871}^{\pm0.030}$ |

Table 1: The quantitative results of text-to-motion on the HumanML3D subset of Motion-X dataset (Lin et al., 2023a), following the unified SMPL-X representation (Bian et al., 2025).

masked transformer. Crucially, in contrast to the original design, we introduce cross-attention layers within the transformer to explicitly model interactions between the control signals and the motion sequence. This modification aims to produce more precise, fine-grained control representations, thereby enhancing both the quality and controllability of the generated motions.

### 3.6 INFERENCE

The inference process starts with an all-masked sequence. We introduce mask tokens at the corresponding positions in the sequence. This enables the autoregressive model to iteratively predict the masked latent signals conditioned on the observed context.

When performing speech-to-gesture and music-to-dance tasks, the speech and music modalities are processed through cross-attention mechanisms within the transformer, interacting with the textual features. This allows the model to generate more fine-grained conditional signals that are temporally aligned with the text. The predicted latents are then fed into the DiT to refine and generate the final motion sequence. The sampling process can be denoted as:

$$\mathbf{x}_{t-1}^i = \frac{1}{\sqrt{\alpha_t}}\left(\mathbf{x}_t^i - \frac{\sqrt{1-\alpha_t}}{\sqrt{1-\bar{\alpha}_t}}\epsilon_\theta(\mathbf{x}_t^i \mid t + \mathbf{z}^i)\right) + \sigma_t\epsilon_t, \tag{6}$$

where $\epsilon_t \sim \mathcal{N}(0, I)$, and $\mathbf{z}^i$ denotes the condition output from the transformer. We adopt classifier-free guidance (CFG) (Chang et al., 2023) to condition the transformer on signal embeddings. At inference time, CFG is applied at the final linear projection layer preceding the softmax operation. At this point, the final logits $l_f$ are computed by adjusting the conditional logits $l_c$ relative to the unconditional logits $l_{uc}$, using a guidance scale $\alpha$:

$$l_f = (1 + \alpha) \cdot l_c - \alpha \cdot l_{uc} \tag{7}$$

| Method | FID$_H$ ↓ | FID$_B$ ↓ | Face L2 Loss ↓ | Beat Align Score ↑ | Diversity ↑ |
|---|---|---|---|---|---|
| Talkshow (Yi et al., 2023) | 26.713 | 74.824 | **7.791** | 6.947 | 13.472 |
| EMAGE (Liu et al., 2024a) | 39.094 | 90.762 | 7.680 | 7.727 | 13.065 |
| MCM (Ling et al., 2023) | 23.946 | 71.241 | 16.983 | 7.993 | 13.167 |
| MotionCraft (Bian et al., 2025) | 18.486 | 27.023 | 10.097 | 8.098 | 10.334 |
| Ours | **17.651** | **25.923** | 9.883 | **8.377** | **14.703** |

Table 2: Results of speech-based motion generation on the BEAT2 dataset (Liu et al., 2024a), following the unified SMPL-X representation (Bian et al., 2025).

## 4 EXPERIMENTS

### 4.1 EXPERIMENT SETTINGS

**Implementation Details.** Our model is implemented on one NVIDIA V100 GPU using PyTorch. For our method, the autoencoder employs a ResNet-based (He et al., 2016) three-layer encoder-decoder architecture with a hidden dimension of 512 and an overall downsampling rate of 4. For the generation process, we utilize a four-layer AdaLN-zero transformer encoder as the masked autoregressive transformer, featuring a hidden dimension of 1024 and 16 attention heads. The diffusion model consists of 4 layers of DiT, where each transformer block has a hidden dimension of 1792 and 8 attention heads. We adopt the AdamW optimizer ($\beta_1 = 0.9, \beta_2 = 0.99$). For training the autoencoder on the HumanML subset of Motion-X, we use a batch size of 256 and a maximum sequence length of 64 frames. For the text-to-motion task, the batch size is set to 50 with a maximum sequence length of 196 frames. The learning rate is initialized at 0.0002 with a linear warmup over 2000 steps. The autoencoder is trained for 50 epochs, while the text-to-motion task is trained for 1500 epochs. During multimodal generation, we first initialize the model with pretrained weights from the text-to-motion autoencoder and fine-tune it on task-specific datasets. Subsequently, we freeze the parameters of the text-to-motion DiT and only fine-tune the masked transformer along with newly incorporated cross-attention layers. The learning rate and training schedule remain consistent with the text-to-motion task. For all three tasks, we employ exponential moving average (EMA) to update model parameters, ensuring training stability. During inference, the classifier-free guidance (CFG) scale is set to 4.5 for text-to-motion, while other tasks use a CFG scale of 6.5.

**Datasets and Metrics.** For the evaluation, we utilize three datasets: HumanML3D (Guo et al., 2022a) for text-to-motion, BEAT2 (Liu et al., 2024a) for speech-to-gesture, and FineDance (Li et al., 2023) for music-to-dance, all following the unified SMPL-X representation (Bian et al., 2025). Regarding the metrics, we use FID, R-Precision, and MM-Dist for text-based motion generation, use FID$_H$, FID$_B$, Face L2 loss, Beat Alignment Score, Diversity for speech-based motion generation, and use FID$_H$, FID$_B$, Diversity for music-based motion generation, respectively. For the detailed explanation of the datasets and metrics, please refer to the appendix.

### 4.2 EVALUATION

**Text-based motion generation.** We conduct an evaluation of our model against prior text-to-motion approaches, including both discrete-domain and continuous-domain methods. As summarized in Table 1, our method clearly surpasses prior techniques on the HumanML subset of Motion-X dataset. Remarkably, our model achieves improvements of 19.3%, 13.5%,

| Method | FID$_H$ ↓ | FID$_B$ ↓ | Div ↑ |
|---|---|---|---|
| Edge (Tseng et al., 2023) | 93.430 | 108.507 | 13.471 |
| Finedance (Li et al., 2023) | 10.747 | 72.229 | 13.813 |
| MCM (Ling et al., 2023) | 4.717 | 78.577 | 14.890 |
| MotionCraft (Bian et al., 2025) | 3.858 | 76.248 | **16.667** |
| Ours | **3.632** | **71.930** | 15.871 |

Table 3: Results of music-based motion generation on the FineDance (Li et al., 2023), following the unified SMPL-X representation (Bian et al., 2025).

and 11.7% in R-Precision for Top-1, 2, 3, respectively. Additionally, we enhance the FID score by 75.2% on this dataset, underscoring the exceptional fidelity of our generated motions. The qualitative results in Figure 4 further support these findings, showing that our approach yields whole-body motions that align closely with the input text.

**Speech-based motion generation.** To assess the speech-driven motion generation, we compare to previous speech-to-gesture methods. Our results, summarized in Table 2, reveal that our method

| Setting | Text-based | | | | Speech-based | |
| | R Precision | | | FID ↓ | $\text{FID}_H$ ↓ | $\text{FID}_B$ ↓ |
| | Top-1 ↑ | Top-2 ↑ | Top-3 ↑ | | | |
| --- | --- | --- | --- | --- | --- | --- |
| Baseline | $0.578^{\pm 0.007}$ | $0.737^{\pm 0.006}$ | $0.787^{\pm 0.005}$ | $9.324^{\pm 0.120}$ | 37.732 | 40.419 |
| + Causal Attention | $0.589^{\pm 0.006}$ | $0.740^{\pm 0.004}$ | $0.798^{\pm 0.006}$ | $9.031^{\pm 0.095}$ | 36.815 | 38.674 |
| + DiT | $0.688^{\pm 0.005}$ | $0.828^{\pm 0.007}$ | $0.851^{\pm 0.004}$ | $5.562^{\pm 0.085}$ | 19.743 | 28.228 |
| + Gated Linear | $0.692^{\pm 0.004}$ | $0.834^{\pm 0.005}$ | $0.877^{\pm 0.006}$ | $4.844^{\pm 0.078}$ | 19.427 | 28.156 |
| + RMSNorm | $0.704^{\pm 0.003}$ | $0.843^{\pm 0.005}$ | $0.898^{\pm 0.005}$ | $4.838^{\pm 0.100}$ | 18.329 | 27.741 |
| + Cross Attention | $0.704^{\pm 0.003}$ | $0.843^{\pm 0.005}$ | $0.898^{\pm 0.005}$ | $4.838^{\pm 0.100}$ | 17.651 | 25.823 |

Table 4: The ablation study of different model components.

achieves good quality and diversity in both hand and body motion generation and excels in aligning with the rhythm of first-person speech. This demonstrates the effectiveness of our framework in motion generation when encompassing different modal signals. However, our method performs worse than single-modal methods. As discussed in (Bian et al., 2025), this is attributed to the random or average expressions in the Motion-X dataset, which confuses the speech-to-gesture training.

**Music-based motion generation.** We further evaluate our framework on the music-to-dance task. As shown in Table 3, our method achieves slightly improved performance over previous approaches, particularly in generating hand motions and body movements.

## 4.3 ABLATION STUDY

**Causal Attention.** To verify the efficacy of the proposed framework, we initially establish a baseline model following the visual MAR setup (Li et al., 2024b), i.e, using the random pseudo reordering for the batched masked prediction, through the bidirectional attention computing. From the results presented in Table 4, we see that the performance of this baseline in the motion area is limited. We attribute this to the difference between human motions and visual images, e.g., the human motion is in a strong temporal sequential structure, in which case a causal attention makes more sense. Therefore, changing the baseline to sequential masked prediction with causal attention improves the performance.

**DiT.** In order to evaluate how the DiTs contribute to the motion quality, we further replace the MLPs in the baseline model with our DiTs. As shown in Table 4, the model generates superior motions with DiTs compared to MLPs, especially in the context of multimodal motion generation. This reveals the superior potential of DiTs in generating motion with complex multimodal contexts.

**Gated Linear Mechanism.** To assess the function of the gated linear mechanism, we ablate this and report the results in Table 4, which indicates that the model outputs motions of higher quality with the inclusion of this mechanism. In the experiments, we observed that the output motions sometimes contain more detailed actions with this mechanism in place.

**RMSNorm.** We also conduct an ablation study to evaluate the function of the RMSNorm and report the results in Table 4. From the results, we see that the model produces better motions when utilizing RMSNorm. In experiments, we found that this module makes the output more stable.

**Cross Attention.** In the baseline model, the multimodal signals are injected with only the AdaLN structure. We then add the cross attention module and observe a significant improvement in multimodal motion generation, as depicted in Table 4.

## 5 CONCLUSION

This paper proposes a new omni motion framework for multimodal whole-body human motion generation. Within this one framework, text, speech, and music signals are all encompassed through AdaLN and cross-attention. The motion generation process is modeled by a continuous masked autoregressive transformer with causal attention, as well as a DiT structure. Extensive experiments have been conducted to verify the efficacy of the proposed framework in different-modality tasks.

**Limitations.** Due to the restricted dataset, the naturalness and generalizability of the motion generation model are still limited, especially in speech and music-driven motion generation.

## 6 ETHICS STATEMENT

This work focuses on the generation of human motion sequences and does not involve interaction with physical human subjects or the use of personal identifiable data. All motion data used for training and evaluation are sourced from publicly available, authorized human motion capture databases (e.g., HumanML3D, KIT-ML). We recognize the responsibility that comes with developing technology capable of generating human-like behavior. Our work is intended for positive applications, such as assisting animation, facilitating physical rehabilitation research, and advancing human-robot interaction. We have thoroughly considered potential misuse, including the generation of biased or harmful actions, and have implemented filtering mechanisms in our data processing pipeline to mitigate such risks. The generative model itself contains no explicit demographic or identity markers, aiming to produce neutral and equitable motion content. We are committed to the ethical development of generative models and will open-source our code to foster transparent research in this domain. There are no conflicts of interest to declare.

## 7 REPRODUCIBILITY STATEMENT

To facilitate the replication of our motion generation framework, we have provided extensive methodological details throughout the manuscript. The motion data utilized is publicly accessible, and we have included demo videos in the supplementary material to qualitatively demonstrate the generated results. We are committed to promoting open science and, upon acceptance, will release the full source code and pre-trained models to the public to ensure the results can be faithfully reproduced and built upon by the research community.

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

# A APPENDIX

## A.1 TRAINING DETAILS

**Encoding of Speech and Music**   Our speech and music encoders are designed to extract temporally aligned, high-level features from raw audio signals for effective speech-to-gesture and music-to-dance generation. The architecture builds upon a multi-layer 1D convolutional network with strided convolutions and leaky ReLU activations. Each convolutional block consists of a series of a block unit that progressively downsample the input waveform while increasing feature dimensionality. The input audio sequence is first processed through multiple stages of temporal aggregation and non-linear transformation, resulting in a sequence of compact and expressive latent representations, whereas the input music typically retains sufficient temporal structure and spectral richness in its raw form for effective motion synthesis. These latent codes capture prosodic, rhythmic, and semantic-like patterns in speech and music, which are then projected into the condition latent space of dimensionality. The final encoder output is transposed to align with the temporal structure expected by the diffusion model, enabling fine-grained cross-modal interaction between speech and motion sequences during generation.

**Training of AE**   We first pretrain a baseline autoencoder on the text-to-motion task. When fine-tuning it on the speech-to-gesture and music-to-dance tasks, the decoder fails to reconstruct valid motion sequences due to discrepancies in data distribution. However, fine-tuning the autoencoder using the reconstruction objective during the multi-modal training incurs high computational costs. Therefore, we independently fine-tune the baseline AE on each dataset using the reconstruction task before multi-modal generation, and employ the resulting models for downstream tasks.

**Motion Representation**   Following previous work Bian et al. (2025), we utilize SMPL-X formatted motion data with an input dimension of (frame length × 322). The parameter structure is organized as follows: Root orientation (0:3): controls global body rotation; body pose (3:66): Governs major body joint rotations; hand articulation (66:156): controls finger movements; jaw pose (156:159): manages mouth opening/closing; facial expression (159:209): drives emotional expressions; facial shape (209: 309): determines static facial structure; translation (309:312): controls global body position; betas (312: 322): represents static body shape parameters. And the maximum motion length is 196. The model's output maintains identical dimensionality (frame length × 322) to ensure full reconstruction capability. This comprehensive parameterization enables simultaneous control of body motion, facial animation, and global positioning within a unified framework.

## A.2 DATASETS.

For text-based motion generation, we evaluate our method on the HumanML3D (Guo et al., 2022a) dataset, which consists of 14,616 high-quality human motions paired with 44,970 text descriptions. The original body-only SMPL (Loper et al., 2023) format of this dataset is extended to whole-body SMPL-X (Pavlakos et al., 2019) format in MotionCraft (Bian et al., 2025), which we follow in the experiments for evaluation. For speech-based motion generation, we evaluate on the BEAT2 dataset (Liu et al., 2024a), which collects 76 hours of data from 30 speakers, standardized into a mesh representation with paired audio and text lines. The motion of the unified SMPL-X format is also extracted (Bian et al., 2025) for multimodal evaluation. For music-based motion generation, the largest dataset FineDance (Li et al., 2023) is utilized for evaluation. This dataset collects dances of 14.6 hours across 22 genres and provides detailed human motions using the SMPL-H format, which is then converted to the unified SMPL-X format and appended by text descriptions.

To enable full-body, multimodal control over motion generation, we convert all datasets to the SMPL-X format. This involves filling in missing facial expressions in HumanML3D and FineDance using average expression coefficients from the training set, as well as transforming the SMPL-H Rot-6D representation in FineDance into axis-angle format via Gram-Schmidt orthogonalization. This conversion achieves better alignment with SMPL-X parameters and introduces minimal errors compared to the official body-retargeting method, while also offering improved computational efficiency.

| Method | R Precision | | | FID ↓ | MM Dist↓ | Div → |
|---|---|---|---|---|---|---|
| | Top-1 ↑ | Top-2 ↑ | Top-3 ↑ | | | |
| GT | $0.511^{\pm0.003}$ | $0.703^{\pm0.003}$ | $0.797^{\pm0.002}$ | $0.002^{\pm0.000}$ | $9.503^{\pm0.065}$ | $2.974^{\pm0.008}$ |
| MDM(Tevet et al., 2023) | $0.418^{\pm0.005}$ | $0.604^{\pm0.005}$ | $0.707^{\pm0.004}$ | $0.489^{\pm0.025}$ | $9.450^{\pm0.066}$ | $3.630^{\pm0.023}$ |
| MotionDiffuse(Zhang et al., 2024b) | $0.491^{\pm0.001}$ | $0.681^{\pm0.001}$ | $0.782^{\pm0.001}$ | $0.630^{\pm0.001}$ | $9.410^{\pm0.049}$ | $3.113^{\pm0.001}$ |
| FineMoGen(Zhang et al., 2023c) | $0.504^{\pm0.002}$ | $0.690^{\pm0.002}$ | $0.784^{\pm0.002}$ | $0.151^{\pm0.008}$ | $9.263^{\pm0.094}$ | $2.998^{\pm0.008}$ |
| Motion-Verse(Zhang et al., 2024c) | $0.496^{\pm0.002}$ | $0.685^{\pm0.002}$ | $0.785^{\pm0.002}$ | $0.415^{\pm0.002}$ | $9.176^{\pm0.074}$ | $3.087^{\pm0.012}$ |
| MCM(Ling et al., 2023) | $0.494^{\pm0.003}$ | $0.682^{\pm0.005}$ | $0.777^{\pm0.003}$ | $0.075^{\pm0.003}$ | $9.484^{\pm0.074}$ | $3.086^{\pm0.011}$ |
| MotionCraft (Bian et al., 2025) | $0.501^{\pm0.003}$ | $0.697^{\pm0.003}$ | $0.796^{\pm0.002}$ | $0.173^{\pm0.002}$ | $9.543^{\pm0.098}$ | $3.025^{\pm0.008}$ |
| MARDM (Meng et al., 2024) | $0.502^{\pm0.003}$ | $0.691^{\pm0.003}$ | $0.787^{\pm0.002}$ | $0.286^{\pm0.003}$ | $9.470^{\pm0.081}$ | $3.346^{\pm0.007}$ |
| Ours | $0.548^{\pm0.003}$ | $0.743^{\pm0.003}$ | $0.837^{\pm0.002}$ | $0.141^{\pm0.003}$ | $9.537^{\pm0.087}$ | $2.856^{\pm0.008}$ |

Table 5: Results of text-to-motion on the original HumanML3D benchmark.

| Method | R Precision | | | FID ↓ | MM Dist↓ |
|---|---|---|---|---|---|
| | Top-1 ↑ | Top-2 ↑ | Top-3 ↑ | | |
| Ours | $\mathbf{0.704}^{\pm0.003}$ | $0.843^{\pm0.005}$ | $\mathbf{0.898}^{\pm0.005}$ | $\mathbf{4.838}^{\pm0.100}$ | $15.871^{\pm0.030}$ |
| Ours-Finetuned | $0.701^{\pm0.002}$ | $\mathbf{0.846}^{\pm0.005}$ | $\mathbf{0.898}^{\pm0.005}$ | $4.843^{\pm0.102}$ | $\mathbf{15.868}^{\pm0.027}$ |

Table 6: Results of text-to-motion after fine-tuning. (On the HumanML3D subset of Motion-X dataset, following the unified SMPL-X representation.)

To ensure consistency with MotionCraft (Bian et al., 2025), we utilize the pretrained motion encoder and text encode, enabling a unified evaluation of the SMPL-X motion representation across different modalities. For datasets that lack corresponding textual annotations—namely FineDance and BEAT2—we generate pseudo-captions such as "A dancer is performing a street dance in the Jazz style to the rhythm of the wildfire" and "A person is giving a speech, and the content is ...", respectively, to support cross-modal learning.

### A.3 METRICS

**Text-based Motion Generation** To assess the quality of the motions generated based on texts compared to the true data, we utilize the Frechet Inception Distance (FID) to evaluate the distribution differences between the generated motions and the ground truth. Additionally, R-Precision is employed to determine how frequently the most relevant motions, identified as top-k closest matches, align with their respective captions within a batch of 32 samples. Lastly, Multi-Modal Distance (MM Dist) is employed to gauge the average Euclidean distance between motion representations and their corresponding textual features.

**Speech-based Motion Generation** For evaluating the quality and diversity of the motions generated based on speech, we employ $FID_H$, $FID_B$, and Diversity metrics. $FID_H$ measures the difference between hand motion distribution and the true gesture distribution, whereas $FID_B$ assesses the divergence between the whole-body motion distributions. The Beat Alignment Score (Li et al., 2021) is used to measure the synchronization between motions and speech beats. To quantify the difference between generated expressions and actual expressions, we use the L2 Loss.

**Music-based Motion Generation** Mirroring the approach used for speech-driven gesture generation, we apply $FID_H$, $FID_B$, and Diversity metrics to evaluate the quality and diversity of

| Method | R Precision | | | FID ↓ | MM Dist↓ |
|---|---|---|---|---|---|
| | Top-1 ↑ | Top-2 ↑ | Top-3 ↑ | | |
| GT | $0.663^{\pm0.006}$ | $0.807^{\pm0.002}$ | $0.864^{\pm0.002}$ | $0.000^{\pm0.000}$ | $15.567^{\pm0.036}$ |
| MotionCraft-Basic (Bian et al., 2025) | $0.590^{\pm0.003}$ | $0.743^{\pm0.004}$ | $0.804^{\pm0.004}$ | $8.477^{\pm0.102}$ | $16.252^{\pm0.035}$ |
| MotionCraft-Mix (Bian et al., 2025) | $0.600^{\pm0.003}$ | $0.747^{\pm0.004}$ | $0.812^{\pm0.006}$ | $6.707^{\pm0.081}$ | $16.334^{\pm0.059}$ |
| Ours-Basic | $0.704^{\pm0.003}$ | $0.843^{\pm0.005}$ | $0.898^{\pm0.005}$ | $4.838^{\pm0.100}$ | $15.871^{\pm0.030}$ |
| Ours-Mix | $\mathbf{0.712}^{\pm0.003}$ | $\mathbf{0.849}^{\pm0.005}$ | $\mathbf{0.904}^{\pm0.004}$ | $\mathbf{4.759}^{\pm0.102}$ | $\mathbf{15.765}^{\pm0.026}$ |

Table 7: Results of text-driven motion generation on the HumanML3D dataset following the mix training setup (Bian et al., 2025).

| Method | FID$_H$ ↓ | FID$_B$ ↓ | Face L2 Loss ↓ | Beat Align Score ↑ | Diversity ↑ |
|---|---|---|---|---|---|
| MotionCraft-Basic (Bian et al., 2025) | 18.486 | 27.023 | 10.097 | 8.098 | 10.334 |
| MotionCraft-Mix (Bian et al., 2025) | 12.882 | 25.187 | **8.906** | 8.226 | 12.595 |
| Ours-Basic | 17.651 | 25.923 | 9.883 | 8.377 | 14.703 |
| Ours-Mix | **12.201** | **25.644** | 8.947 | **8.430** | **15.003** |

Table 8: Results of speech-driven motion generation on the BEAT2 dataset (Liu et al., 2024a) following the mix training setup (Bian et al., 2025).

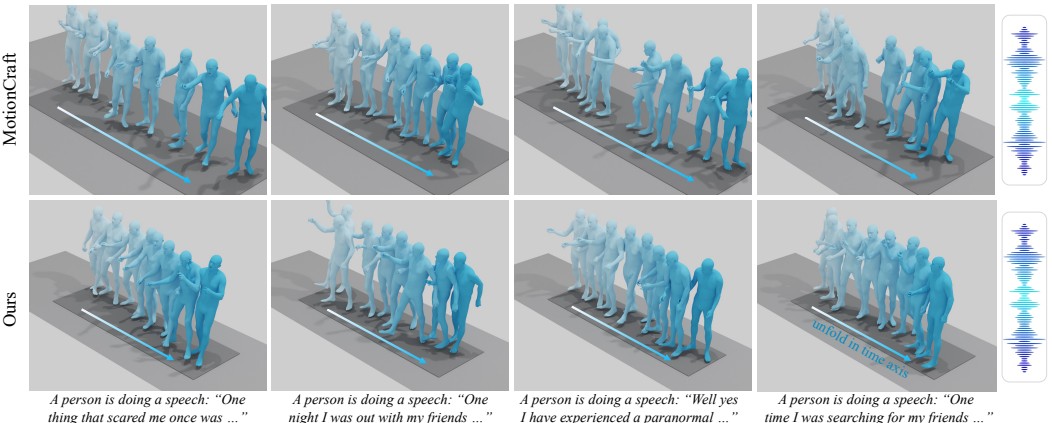

A person is doing a speech: "One thing that scared me once was ..."  A person is doing a speech: "One night I was out with my friends ..."  A person is doing a speech: "Well yes I have experienced a paranormal ..."  A person is doing a speech: "One time I was searching for my friends ..."

Figure 5: The qualitative results of speech-driven motion generation.

music-induced hand and whole-body movements. This approach ensures that the generated motions exhibit both high fidelity and variation.

### A.4 ADDITIONAL EXPERIMENT RESULTS

**More Results on Text-to-motion** To provide a more comprehensive evaluation, we conduct additional comparisons on the original HumanML3D benchmark using the body-only H3D format, which contains redundant motion information. Here we mainly compare with the methods without VQ-VAE. As shown in Tab. 5, OmniMotion consistently outperforms these baselines in terms of text-motion alignment, motion quality, and diversity, demonstrating its superior generalization capability across different motion representations.

**Text-to-motion Evaluation after fine-tuning** We conduct a comprehensive evaluation on the final model (after fine-tuning on both speech-to-gesture and music-to-dance datasets) and present the results below. Since textual conditioning participates throughout the entire training pipeline, our model does not suffer from catastrophic forgetting after fine-tuning. This confirms the robustness of our architecture's knowledge retention capabilities under different training paradigms.

**Evaluation of OmniMotion Variants** Following the same strategy as MotionCraft (Bian et al., 2025), we train two variants of our model: OmniMotion-Base and OmniMotion-Mix. OmniMotion-Base is a text-to-motion model pretrained solely on HumanML3D, while OmniMotion-Mix is trained on a combined dataset comprising HumanML3D, BEAT2, and FineDance to enable multimodal motion generation. Quantitative results on the text-to-motion task are summarized in Table 7.

We further evaluate OmniMotion-Mix across all three modalities. For the speech-to-gesture and music-to-dance tasks, we fine-tune the model on the respective target datasets. The corresponding results are reported in Table 8 and Table 9, respectively.

### A.5 MORE VISUALIZATION

We display more visual results of speech-driven and music-driven motion generation in Figure 5 and Figure 6, respectively.

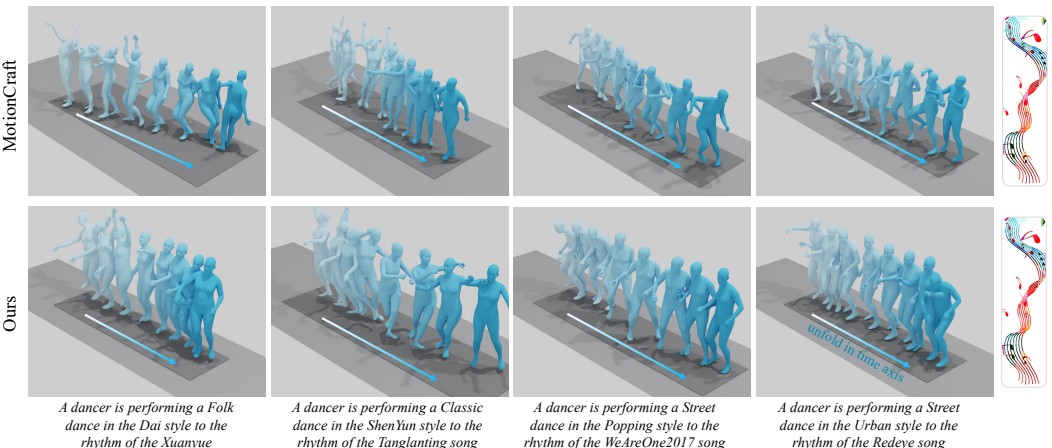

Figure 6: The qualitative results of music-driven motion generation.

| Method | FID$_H$ ↓ | FID$_B$ ↓ | Div ↑ |
|---|---|---|---|
| MotionCraft-Basic (Bian et al., 2025) | 3.858 | 76.248 | 16.667 |
| MotionCraft-Mix (Bian et al., 2025) | 2.849 | 67.159 | **18.483** |
| Ours-Basic | 3.632 | 71.930 | 15.871 |
| Ours-Mix | **2.781** | **64.380** | 17.605 |

Table 9: Results of music-driven motion generation on the FineDance dataset (Li et al., 2023) following the mix training setup (Bian et al., 2025).

## B  THE USE OF LARGE LANGUAGE MODELS (LLMs)

LLMs were used in the preparation of this work solely for the purpose of grammar correction and proofreading. The LLM did not contribute to the ideation, scientific content, writing, or analysis of the research. The authors take full responsibility for the entire intellectual content of this paper.

