# OpenReview forum: "OmniMotion: Multimodal Motion Generation with Continuous Masked Autoregression"
_ICLR.cc/2026/Conference — ICLR 2026 Conference Withdrawn Submission_

### Official Review · Reviewer_hmuD · 2025-10-28

**Soundness:** 3
**Presentation:** 2
**Contribution:** 2
**Rating:** 4
**Confidence:** 4

**Summary:**

This paper addresses multimodal human motion generation, focusing on improving the quality of generated motions and the generalization ability across different input modalities.
To this end, the authors propose a masked autoregressive motion transformer with causal attention, enhanced by Gated Linear Attention and RMSNorm modules.
In addition, they adopt a Diffusion Transformer (DiT) architecture with AdaLN and cross-attention–based conditioning mechanisms.
Experimental results demonstrate consistent performance improvements across multiple tasks.

**Strengths:**

1. The paper is clearly written and logically structured. The experimental ablations are comprehensive, allowing readers to easily understand the differences and improvements over prior work.

2. The proposed task—multimodal conditional motion generation—is both challenging and promising. The idea of leveraging data from different tasks to improve generalization in motion generation is novel and relevant to the community.

**Weaknesses:**

1. Although the paper claims that using data from other tasks improves the model’s performance across tasks, Table 6 in the appendix only shows results for the text-to-motion task before and after fine-tuning. The results do not clearly demonstrate cross-task improvement, and the metric changes are marginal. Could this indicate that the claimed multi-task benefit may not actually occur in practice?

2. Several proposed architectural modifications seem to be engineering-level improvements rather than conceptual innovations. For instance, replacing bidirectional attention with masked causal attention, predicting one token at a time instead of in batches, and adding Gated Linear and RMSNorm modules—all follow currently popular architectural trends. Hence, presenting these as key contributions might be somewhat limited in novelty.

3. The authors argue that VQ-based methods introduce quantization errors, motivating their choice of an autoencoder (AE)-based design. However, recent VQ-based models such as MoMask and LAMP have shown strong performance. It would strengthen the paper if the authors could report direct comparisons or quantitative results showing why non-quantized (AE-based) models are superior in the autoregressive generation setting.

**Questions:**

1. The paper mentions that adopting DiT increases computational cost. Could the authors provide quantitative inference-time comparisons between the DiT and the original MLP-based denoising network?

2. Have the authors experimented with alternative conditioning injection methods, or did they directly choose AdaLN and cross-attention? If the latter, could they elaborate on why these particular mechanisms were selected, especially since the paper highlights them as a contribution?

3. In the VAE component, how does the proposed design differ from that used in MARDM?

4. The paper claims that the Gated Linear Mechanism reduces the attention paid to redundant frames while emphasizing relevant ones. Is there any direct experimental evidence supporting this claim?

5. Could the authors provide a detailed explanation of why DiT is more advantageous than MLPs or convolution-based denoisers (as used in Stable Diffusion) for motion generation?

6. In Table 4, only text-based and speech-based settings are reported. Is there a reason why music-based ablation experiments are missing?

---

### Official Review · Reviewer_dqfX · 2025-10-31

**Soundness:** 3
**Presentation:** 3
**Contribution:** 2
**Rating:** 2
**Confidence:** 5

**Summary:**

This paper presents a framework for multi-modal motion generation. The method is simple and easy to follow, and the paper is overall well written with clean figures and tables. However, despite the good presentation, the technical novelty and empirical validation appear limited, and several claims seem overstated relative to the actual contributions and results.

**Strengths:**

The proposed method is simple and easy to understand, which makes the paper accessible.

The writing quality, figures, and tables are clear and formatted.

**Weaknesses:**

- The title and scope feel overstated. The work focuses on a limited number of modalities, yet it is framed as “omni-,” which seems misleading. More modalities could be scene environments, other human motion, etc.

- The demo results are weak: generated motions often appear frozen or unnatural, and foot sliding is severe in many cases (the video provided in the .ppt file).

- The paper omits many related baselines. Given the large number of existing motion generation methods, this omission makes it difficult to assess the true contribution.

- The ablation study lacks visualizations or demo comparisons. From the reported results in Table 4, if I understand correctly (each row represents the baseline plus all previous modules), switching to DiT brings the largest improvement, but this is conceptually a rather smallest technical contribution. Other components (stated as contributions), such as the Gated Linear and RMSNorm modules, lead to only marginal gains, which further weakens the overall claim of the paper.

- The provided demos are limited. There are no results for music-to-motion.

**Questions:**

For Table 4, if I understand correctly, it seems that each row represents the baseline plus all the previous modules, but this is not clearly explained in the text. Without a detailed description, readers might mistakenly assume that each row corresponds to the baseline plus only one additional module.

---

### Official Review · Reviewer_zsey · 2025-10-31

**Soundness:** 3
**Presentation:** 3
**Contribution:** 3
**Rating:** 6
**Confidence:** 3

**Summary:**

This paper proposes OmniMotion, a unified framework for whole-body human motion generation across multiple modalities (text, speech, and music). The core contribution is a continuous masked autoregressive (MAR) transformer that employs causal attention to maintain temporal coherence, avoiding the quantization errors inherent in VQ-VAE-based discrete methods. The framework incorporates: (1) a gated linear attention mechanism for adaptive feature selection, (2) RMSNorm for training stability across heterogeneous modalities, and (3) Diffusion Transformer (DiT) blocks for refinement. The model is pretrained on text-to-motion data and then fine-tuned for speech-to-gesture and music-to-dance tasks. Experiments on HumanML3D, BEAT2, and FineDance datasets demonstrate improvements over existing methods.

**Strengths:**

1. The use of causal attention in MAR for motion generation is sensible given the sequential nature of motion, distinguishing this work from image-based MAR methods.
2. The paper provides extensive experiments across three different modalities with appropriate metrics and baselines, including thorough ablation studies.
3.  Using SMPL-X for whole-body motion across datasets is practical and enables fair comparison. This task is novel and interesting from my point of view.

**Weaknesses:**

1. The paper is primarily an engineering effort combining existing techniques (MAR, DiT, AdaLN, cross-attention).
2. The paper might somewhat overclaim the "omni" capabilities. The paper claims multimodal training allows "each modality to benefit from patterns in other modalities", but the frozen DiT and separate fine-tuning directly contradict this. Speech and music don't actually influence the core generative model. This is more similar to adaptation, not unification.
3. Presentation issues:
- Dense writing in technical sections makes it hard to follow.

**Questions:**

1. Have you tried joint training on all three modalities simultaneously (without freezing)? This would be the actual "omni" approach claimed in the title.
2. Can you provide evidence that training on music improves text-to-motion, or that speech data helps music-to-dance? Ablations training on subsets would demonstrate actual knowledge sharing.

---

### Official Review · Reviewer_XUWK · 2025-10-31

**Soundness:** 1
**Presentation:** 3
**Contribution:** 2
**Rating:** 2
**Confidence:** 5

**Summary:**

This paper proposes Omnimotion, a framework for multimodal whole-body human motion generation, including text-to-motion, music-to-dance and speech-to-gesture. The usage of AdaLN, RMSNorm proves that these modules is well-adapted to multimodal-conditioned motion generation frameworks. The experiments demonstrate that OmniMotion performs well on several benchmarks.

**Strengths:**

This paper proposes Omnimotion, a framework for multimodal whole-body human motion generation, including text-to-motion, music-to-dance and speech-to-gesture. This framework performs well on FineDance and BEAT2 dataset.

**Weaknesses:**

1. The motivation of this paper is not clear. The author mentioned that previous works focuses on single-conditioned motion generation, and many single-conditioned motion generation methods (VQ-based and continuous) suffer from their respective outcome. However, it is not well stated the relationship with multiple condition motion generation. It seems that this paper aims to tackle multimodal-conditioned motion generation, but a large proportion of Introduction describe the shortcoming of current text-to-motion methods’architecture. Overall, motivation is confusing.
2. It seems that OmniMotion is an directly application of MAR on motion generation.
3. HumanML3D is the main and basic benchmark for text-to-motion generation. However, OmniMotion performs not well on HumanML3D.
4. Experiments is not sufficient. (1) After adapting to other conditions, will the performances on text-to-motion get worse? (2) Why this paper apply pretrain then finetune/adapt paradigm for other conditions beyond text?  How does the network perform if we directly trained on speech and music?
5. MARDM is the first to apply MAR-based methods on text-to-motion generation, but it seems performs worse than the so-called suboptimal VQ-based methods (T2M-GPT, MoMask) when using the 263-dim H3D format. This paper follows the same MAR-like paradigm, but it seems that the author didn't dig into this.
6. MAR apply a lightweight MLP diffusion head, while OmniMotion apply a 4 layers transformer. Supposed that a motion sequence with 200 frames, for each frame pose, the 4 layers transformer needs to run the whole denoising process for 200 times. This is a huge computational overhead, which may lead to OmniMotion less competitive than the previous VQ-based methods on both motion quality and computational cost.

**Questions:**

1.	Lacking comparisons of many SOTA works in Year 2025 dedicated for text-to-motion in HumanML3D in Table 5. In addition, MM Dist and Diversity has the wrong place.
2.	Should the DiT be frozen when injecting additional conditions according to Section 3.5?  DiT is either frozen or trained in Figure 2(c). This is confusing.
3.	What is Face L2 loss?  The author didn't introduce it or cite the reference paper.
4.	In line 445, “However, our method performs worse than single-modal methods. ”What is the meaning of this sentence? From Table 2, it seems that the proposed method is overall better than single-modal methods ( EMAGE, TalkSHOW). The overall statement should be further checked.
5.	In line 895, “Here we mainly compare with the methods without VQ-VAE.” This paper apply MAR to design a motion generation framework and aims to tackle the shortcoming of VQ-based methods (i.e., the inevitable quantization error), as claimed in Introduction. Thus, it is not reasonable to avoid comparison with VQ-based methods.
6.	Table 6 is given but not referred in this paper.
7.	The results of EMAGE in Table 2 is totally different with those in EMAGE paper. It would be better if the authors clarify during testing how many speakers are used and if the split of test set is same as the original BEAT2 dataset.

---

### Note · Authors · 2025-11-13

I have read and agree with the venue's withdrawal policy on behalf of myself and my co-authors.